# The Roles of Antisense Long Noncoding RNAs in Tumorigenesis and Development through Cis-Regulation of Neighbouring Genes

**DOI:** 10.3390/biom13040684

**Published:** 2023-04-18

**Authors:** Binyuan Jiang, Yeqin Yuan, Ting Yi, Wei Dang

**Affiliations:** 1Department of Clinical Laboratory, The Affiliated Changsha Central Hospital, Hengyang Medical School, University of South China, Changsha 410004, China; 2Medical Research Center, The Affiliated Changsha Central Hospital, Hengyang Medical School, University of South China, Changsha 410004, China; 3Department of Science and Education, The Affiliated Changsha Central Hospital, Hengyang Medical School, University of South China, Changsha 410004, China

**Keywords:** antisense long noncoding RNA, neighbouring gene, cis-regulation, tumour

## Abstract

Antisense long noncoding RNA (as-lncRNA) is a lncRNA transcribed in reverse orientation that is partially or completely complementary to the corresponding sense protein-coding or noncoding genes. As-lncRNAs, one of the natural antisense transcripts (NATs), can regulate the expression of their adjacent sense genes through a variety of mechanisms, affect the biological activities of cells, and further participate in the occurrence and development of a variety of tumours. This study explores the functional roles of as-lncRNAs, which can cis-regulate protein-coding sense genes, in tumour aetiology to understand the occurrence and development of malignant tumours in depth and provide a better theoretical basis for tumour therapy targeting lncRNAs.

## 1. Introduction

Long noncoding RNAs (lncRNAs) are defined as nonprotein-coding transcripts with more than 200 nucleotides [1]. Antisense long noncoding RNA (as-lncRNA) are a special kind of lncRNA belonging to a natural antisense transcript (NAT), which is the opposite to the transcription direction of protein-encoded transcripts and widely exists in eukaryotes [2]. A large proportion of lncRNAs are classified as antisense lncRNAs [3,4]. As-lncRNAs generally overlap with the coding region, promoter region, or other regulatory region of their neighbouring sense genes, and can achieve a wide range of biological and cytological functions through cis or trans regulation of neighbouring sense genes [5]. as-lncRNAs have been found to regulate the expression of their neighbouring sense genes at multiple levels through DNA–RNA, RNA–RNA, or protein–RNA interactions in the nucleus or cytoplasm, including pretranscriptional, transcriptional, posttranscriptional, translational, and posttranslational levels [6,7]. Abnormal expression of as-lncRNAs is associated with the pathogenesis of many diseases, such as cancer and neurological diseases [8]. This paper focuses on the function and mechanism of as-lncRNAs affecting tumorigenesis and development by regulating the expression of sense genes.

## 2. Classification of Antisense lncRNAs

According to the mode of action of antisense lncRNAs, as-lncRNAs can be divided into cis-acting as-lncRNAs and trans-acting as-lncRNAs [9]. Cis-acting as-lncRNAs are lncRNAs that affect gene expression or chromatin status at the same loci. Trans-acting as-lncRNAs can interact with genes located at distant loci or even on other chromosomes to play a regulatory role [9,10].

### 2.1. Cis- Acting as-lncRNAs

Cis-acting as-lncRNAs can be further classified according to their proximity to corresponding neighbouring genes (Figure 1), which can be divided into head-to-head as-lncRNAs, i.e., neighbouring genes where the 5′ ends of as-lncRNAs are aligned together and are called head-to-head as-lncRNAs or reverse as-lncRNAs. Tail-to-tail as-lncRNAs are neighbouring genes that are aligned with the 3′ end of as-lncRNAs and are called tail-to-tail as-lncRNAs or convergent as-lncRNAs [11]. Nearby as-lncRNAs in a head-to-head manner, that is, the 5′ end of neighbouring genes is close to the 5′ end of as-lncRNAs, and nearby as-lncRNAs in a tail-to-tail manner, that is, the 3′ end of neighbouring genes is close to the 3′ end of as-lncRNAs. These two forms of as-lncRNAs are also known as intergenic as-lncRNAs [11,12]. Fully overlapping as-lncRNAs, that is, as-lncRNAs completely overlap with neighbouring genes [13,14,15].

### 2.2. Trans-Acting as-lncRNAs

In contrast to cis-acting as-lncRNAs, trans-acting as-lncRNAs mainly affect chromatin status and gene expression at distant sites. Kong et al. found that ZFPM2-AS1 stabilizes the expression of MIF and inhibits the p53 signaling pathway in a trans-acting manner, thereby promoting the progression of gastric cancer [16]. The expression of the neighbouring genes ABRA, ANGPT1, LRP12, OXR1, and ZFPM2 was not affected by ZFPM2-AS1. Therefore, by using sodium dodecyl sulfate-polyacrylamide gel electrophoretic silver staining and mass spectrometry analysis, they identified that ZFPM2-AS1 binds MIF and protects the stability of its protein, thereby promoting the proliferation of gastric cancer cells and inhibiting the apoptosis of gastric cancer cells by reducing the nuclear localization of p53 [16]. In addition, as-lncRNAs transregulate the expression of target genes through the ceRNA (competing endogenous RNA) mechanism. Wang et al. found that HOXD-AS1 was mainly located in the cytoplasm and shared a miR-130a-3p response element with SOX4, thus preventing the degradation of SOX4 mediated by miR-130a-3p [17]. Shuai et al. reported that MNX1-AS1 sponged miR-6785-5p and upregulated the expression of Bcl-2 in gastric cancer cells. Meanwhile, lncRNA MNX1-AS1 inhibited BTG2 expression through EZH2-induced H3K27me3 modification in the BTG2 promoter region [18]. Table 1 summarized the regulation of trans-acting as-lncRNAs on target genes and their effects on tumor progression.

## 3. Regulation Mechanism of as-lncRNAs on Neighboring Genes

Compared with protein encoding genes, the average expression level of most lncRNAs is 10 times lower and generally has higher cell and tissue specificity [19,20]. Some epigenetic modifications encompass DNA methylations of cytosine in CpG islands lead to downregulation of as-lncRNA expression [21]. ZNF582-AS1 is downregulated in clear cell renal cell carcinoma involving DNA methylation at the CpG islands within its promoter [22]. Although lncRNAs exhibit this low expression pattern, their biological role is still considered important because increasing evidence has shown that lncRNA mutations and disorders are associated with a variety of human diseases [23]. LncRNAs are capable of tuning gene expression and impacting cellular signaling cascades in the cancer landscape [24]. As a kind of lncRNAs, as-lncRNAs can also play an important functional role in disease through different regulatory mechanisms [25]. A monitoring study on the half-life of lncRNA showed that its antisense splicing form had higher stability, which led to speculation that as-lncRNAs had multiple molecular functions depending on this biochemical feature [19,26].

The regulatory mechanisms of as-lncRNAs on neighbouring genes are very diverse [6,27], and they can play a role at almost all gene regulation levels [20,28]. At the pretranscriptional level, as-lncRNAs can be used as guides to lead proteins into specific parts of the genome or as decoys to keep proteins away from chromatin and regulate the expression of neighbouring genes through epigenetic changes in target genes caused by histone modification or DNA cytosine methylation [29]. At the transcriptional level, as-lncRNAs regulate the transcription of adjacent target genes by directly affecting the transcriptional activity of target gene promoters [30]. At the posttranscriptional level, as-lncRNAs form RNA–RNA double strands through overlapping regions with neighbouring genes to make neighbouring mRNAs more stable or to regulate the expression of neighbouring genes by changing the alternative splicing of target mRNAs [20,31]. At the translation level, as-lncRNAs regulate the translation of neighbouring genes by influencing the recruitment of target mRNAs to polyribosomes [7]. At the posttranslational level, as-lncRNAs affect the formation of dimers of target proteins, thereby regulating the enzyme activity of neighbouring target proteins [32]. Figure 2 summarizes the mechanism by which as-lncRNAs regulate neighbouring genes (Figure 2).

### 3.1. Regulation of Neighbouring Genes by as-lncRNAs at the Pretranscriptional Level

As-lncRNAs regulate neighbouring genes at the pretranscriptional level, mainly through epigenetic modification affecting their transcription. Zhang et al. reported that as-lncRNA EZR-AS1, on the one hand, increased the transcription of its neighbouring gene EZR by forming a complex with RNA polymerase Ⅱ. On the other hand, EZR-AS1 recruits SMYD3 (lysine methyltransferase) and binds it to the GC-rich SBS-1 site downstream of the EZR promoter, resulting in H3K4 methylation in the EZR promoter region, leading to upregulation of EZR transcription in oesophageal squamous cell carcinoma (ESCC) cells [33]. Similarly, SSTR5-AS1 increases the enrichment of MLL3 and H3K4me3 in the SSTR5 promoter region by interacting with MLL3 and then inducing the transcription of SSTR5 [34]. SEMA3B-AS1 upregulates SEMA3B transcription by increasing MLL4 enrichment and H3K4me3 levels in the SEMA3B promoter region [35]. Among them, MLL3 and MLL4 are members of myeloid/lymphoid or mixed line leukaemia (MLL) family genes and are important enzymes that modify H3K4 methylation [34,36,37]. Yang et al. found that as-lncRNA HOXD-AS1 could recruit the PRC2 (histone methyltransferase) complex to bind to the promoter region of HOXD3, induce inhibition of the accumulation of labelled H3K27me3, and then inhibit the transcription of HOXD3 [38]. ZNF667-AS1 interacts with DNA demethylase (TET1) to convert 5-methylcytosine (5-mC) from the ZNF667 promoter region to 5-hydroxymethylcytosine (5-hmC) to activate the transcription of ZNF667. In addition, ZNF667-AS1 also interacts with histone lysine demethylase (UTX) to reduce the enrichment of H3K27me3 in the ZNF667 promoter region and increase the transcription of ZNF667 [39]. Another as-lncRNA, ID2-AS1, was proven to promote ID2 transcription by blocking the binding of histone deacetylase 8 (HDAC8) to the ID2 enhancer and at the same time increase the accumulation of the gene activation marker H3K27ac in the ID2 enhancer region [40].

### 3.2. Regulation of Neighbouring Genes by as-lncRNAs at the Transcriptional Level

As-lncRNAs regulate neighbouring genes mainly by affecting the transcriptional activity of gene promoter regions directly or by binding with transcription-related factors indirectly at the transcriptional level. as-lncRNA IRF1-AS was found to form a complex with ILF3 (interleukin enhancer binding Factor 3) and DHX9 (Dexh-box helicase 9) in the nucleus of the IRF1 promoter region and activate the transcription of the adjacent gene IRF1. ILF3 and DHX9 are correlated and function as transcriptional coactivators [41]. Another as-lncRNA, TPT1-AS1, was proven to directly induce the transcriptional activity of the TPT1 promoter without inhibiting the degradation of TPT1 mRNA, thus promoting the transcription of the adjacent gene TPT1 [30].

### 3.3. Regulation of Neighbouring Genes by as-lncRNAs at the Posttranscriptional Level

The regulation of as-lncRNAs on adjacent genes at the posttranscriptional level is mainly achieved by affecting the stability of mRNA, participating in the alternative splicing of mRNA, blocking the binding sites of miRNAs, and adsorbing miRNAs as ceRNA sponges.

#### 3.3.1. As-lncRNAs Regulate Neighbouring Genes by Affecting Their mRNA Stability

Although as-lncRNAs are not completely uniform regulatory elements for neighbouring genes, they show some common characteristics, such as sequence complementarity with neighbouring sense genes [2]. Zhang et al. found that the as-lncRNA FOXC2-AS1 shared the overlapping region with its adjacent gene FOXC2 mRNA to form an RNA–RNA double strand, which made FOXC2 mRNA more stable and less susceptible to ribonuclease degradation, thus increasing FOXC2 expression [31]. As-lncRNAs have complementary or overlapping sequences with their adjacent gene mRNAs and are mainly located in the cytoplasm, which makes it possible for them to form RNA–RNA double chains. There are many as-lncRNAs that increase the expression of neighbouring genes by forming RNA–RNA double strands with the mRNA of neighbouring genes, such as IGFBP7-AS1 stabilizing IGFBP7 [42] and FGFR3-AS1 stabilizing FGFR3 mRNA [43]. NR4A1AS stabilizes NR4A1 mRNA [44], Sirt1 antisense lncRNA stabilizes Sirt1 mRNA [45], PDCD4-AS1 stabilizes PDCD4 mRNA [46], and TBX5-AS1:2 stabilizes TBX5 mRNA [47].

In addition to regulating the expression of neighbouring genes by forming RNA–RNA double strands with the mRNA of neighbouring genes, as-lncRNAs also affect the stability of neighbouring gene mRNAs through other mechanisms. LDLRAD4-AS1 directly interacts with LDLRAD4 mRNA, mainly through its 1-1098 bp sequence region, and destroy the stability of LDLRAD4 mRNA [48]. Zhao et al. reported that MACC1-AS1 promotes the stability of MACC1 mRNA by promoting AMPK phosphorylation, leading to the translocation of the RNA-binding protein Lin28 from the nucleus to the cytoplasm [49]. The direct binding of NR4A1AS and NR4A1 mRNA 3′-UTR reduces the binding of UPF1 protein to NR4A1 mRNA, prevents UPF1-mediated mRNA degradation, and makes NR4A1 mRNA more stable [44]. Han et al. found that ZFPM2-AS1, ZFPM2 mRNA, and UFP1 protein form a binding complex, which makes ZFPM2 mRNA unstable [50]. UPF1 has been proven to be an important factor in the degradation of abnormal mRNA [51]. Similarly, PDCD4-AS1 promotes the stability of PDCD4 mRNA by negatively regulating the binding of HuR to PDCD4 mRNA [46]. HuR (Human ANTIGEN R) has been reported to play an important role in destabilizing RNA [52,53]. Through SND1 protein, PTB-AS increased the binding ability of PTB-AS to the PTBP1 mRNA 3′UTR, stabilized PTBP1 mRNA, and maintained the expression of PTBP1 [54]. SND1 was confirmed to have a preference for binding RNA double strands, which stabilized mRNA [55,56].

#### 3.3.2. As-lncRNAs Regulate Adjacent Genes by Changing the Alternative Splicing of mRNA

The as-lncRNA EVA1A-AS inhibits the splicing of EVA1A intron 2 by reducing U2 snRNP (U2-type spliceosome) recruitment to EVA1A pre-mRNA, thereby inhibiting the expression of EVA1A [57]. The MBNL3 splicing factor transcribed exon 4 of the as-lncRNA PXN-AS1 into PXN-AS-L transcripts containing exon 4 through selective splicing to promote the expression of PXN mRNA and protein [58].

#### 3.3.3. As-lncRNAs Regulate Their Neighbouring Genes by Masking the Binding Sites of miRNAs or Adsorbing miRNAs as ceRNA Sponges

The as-lncRNA PTB-AS masks the binding site of miR-9 in the PTBP1-3 ‘UTR, preventing miR-9 from mediating the negative regulation of PTBP1 and maintaining the stability of PTBP1 mRNA [54]. Yuan et al. found that the PXN-AS1-L transcript of lncRNA-PXN-AS1, binding to the 3′UTR of PXN mRNA, covered the binding site of miR-24, protected PXN mRNA from miR-24-induced degradation, and upregulated the expression of PXN mRNA and protein [58]. The as-lncRNA SOCS2-AS1 upregulates SOCS2 expression through competitive adsorption of miR-1264 [59].

### 3.4. As-lncRNAs Regulate Neighbouring Genes at the Translation Level

The as-lncRNA RASSF1-AS1 directly binds RASSF1A mRNA, inhibits translation, and reduces RASSF1A protein levels without affecting RASSF1A mRNA levels [60]. Qi et al. proposed that GABPB1-AS1 inhibits GABPB1 translation by preventing GABPB1 mRNA from assembling polyribosomes and binding with eIF4A [7]. Eukaryotic initiation factor-4A (eIF4A) is required for mRNA binding to the 40S ribosomal subunit during translation [61]. lncRNA pXN-AS1-S transcript without exon 4 prevents binding of PXN mRNA to translation extension factor, inhibits translation extension of PXN mRNA, and downregulates PXN protein expression by binding to the CDS region of PXN mRNA [58].

### 3.5. Regulation of Neighbouring Genes by as-lncRNAs at the Posttranslational Level

The as-lncRNA IDH1-AS1 can promote the homologous dimerization of IDH1 and increase the enzyme activity of IDH1 without changing the expression of IDH1 [32].

## 4. Antisense lncRNAs Affect the Occurrence and Development of Tumours by Regulating Neighbouring Genes

An increasing number of studies have suggested that as-lncRNAs are involved in tumour cell proliferation, migration, invasion, apoptosis, and angiogenesis, and that affect the occurrence and development of tumours by regulating the expression of neighbouring genes [48]. Most of these regulated neighbouring genes are known oncogenic or suppressor genes. Table 2 summarizes the effects of different as-lncRNAs on the occurrence and development of different types of tumours by regulating neighbouring genes (Table 1).

By interacting with TET1 and UTX, ZNF667-AS1 promotes the expression of ZNF667 and E-cadherin, inhibits the proliferation and invasion of oesophageal squamous cell carcinoma, and thus acts as a tumour suppressor gene to inhibit the progression of oesophageal squamous cell carcinoma [39]. Interferon-induced as-lncRNA IRF1-AS promotes the interferon response and inhibits the progression of oesophageal squamous cell carcinoma through a positive regulatory loop with the adjacent tumour suppressor gene IRF1 [41,64]. The as-lncRNA EZR-AS1 promotes the mobility and aggressiveness of oesophageal squamous cell carcinoma cells by enhancing the transcription of the oncogenic gene EZR [33,65,66]. The as-lncRNA ZFPM2-AS1 downregulates the expression of the tumour suppressor gene ZFPM2 [67] through its interaction with UPF1 and promotes the proliferation, invasion, and EMT of lung adenocarcinoma cells, thus promoting the progression of lung adenocarcinoma [50]. TPT1-AS1 promotes the proliferation, invasion, and metastasis of epithelial ovarian cancer by inducing the expression of the oncogenic gene TPT1 [30,68,69]. SOCS2-AS1 inhibits the proliferation and metastasis of colorectal cancer cells and inhibits the progression of colorectal cancer by stabilizing the tumour suppressor gene SOCS2 [70,71]. In colorectal cancer, the as-lncRNA LDLRAD4-AS1 reduces the expression of LDLRAD4 and further upregulates Snail expression by disrupting the stability of LDLRAD4 mRNA, leading to epithelial-mesenchymal transformation (EMT), thereby promoting the metastasis of colorectal cancer [48]. The as-lncRNA HOXD-AS1 inhibits the growth and metastasis of colorectal cancer by inhibiting the transcription of the oncogenic gene HOXD3 [72,73] and further inhibiting the transcriptional activation of integrin β3 and MAPK/AKT signal transduction [38]. The as-lncRNA NR4A1AS promotes the growth and metastasis of colorectal cancer cells by upregulating the expression of the oncogenic gene NR4A1 [44,74]. Yuan et al. reported that splicing factor MBNL3 increased the expression of oncogenic gene PXN [75,76] through selective splicing of lncRNA-PXN-AS1, thereby promoting the occurrence of liver cancer [58]. The as-lncRNA LASP1-AS promotes the proliferation and metastasis of HCC by increasing the expression of its adjacent oncogenic gene LASP1 [77,78]. ID2-AS1 has been found to inhibit HCC metastasis by regulating chromatin modification, promoting the transcription of the adjacent tumour suppressor gene ID2 [79,80,81], and thereby inhibiting the EMT process. The as-lncRNA PTB-AS has been found to significantly promote the occurrence of glioma by pairing with the extended base of the PTBP1 3′UTR with the assistance of SND1 to maintain the level of PTBP1 [54]. PDCD4-AS1 inhibits the proliferation and migration of breast cancer cells by promoting the stability of the tumour suppressor gene PDCD4 [82] mRNA, thus controlling the progression of breast cancer [46]. MACC1-AS1 enhances glycolysis and the antioxidant activity of gastric cancer cells under metabolic stress and promotes the malignant phenotype of gastric cancer cells through the mRNA stability of the oncogenic gene MACC1 [83], mediated by AMPK/Lin28 [49]. FOXF1-AS1 promotes the migration and invasion of osteosarcoma cells by activating the FOXF1/MMP-2/MMP-9 signaling pathway induced by the adjacent oncogenic gene FOXF1 [84,85,86]. The as-lncRNA FGFR3-AS1 has been confirmed to promote the progression of osteosarcoma by positively regulating the expression of its adjacent oncogenic gene FGFR3 [43,87]. ZEB1-AS1 promotes bladder cancer cell migration and invasion by up-regulating the expression of ZEB1. Mechanistically, ZEB1-AS1 activates the translation of ZEB1 mRNA by recruiting AUF1, which is able to bind to (A+U)-rich elements within 3ʹ-untranslated region (3ʹ-UTR) of target mRNA and promote translation without affecting the mRNA level [62]. IDH1-AS1 did not significantly affect the expression of IDH1 mRNA or protein, but it was involved in the enhancement of IDH1 enzyme activity, and it further promoted prostate cancer progression [63].

## 5. Summary and Future Prospects

This review focuses on tumour-related as-lncRNAs and discusses the mechanisms by which these lncRNAs regulate adjacent genes at different regulatory levels and their further impact on tumour progression. A considerable portion of as-lncRNAs cis-regulate the transcription of adjacent protein-coding genes, precisely control the spatiotemporal expression of these gene loci, and participate in developmental and other biological processes related to them. as-lncRNAs are essential for healthy cell metabolism, and their abnormal regulation is believed to be the leading cause of human diseases, including tumours. Therefore, oligonucleotide technology targeting as-lncRNAs has attracted the interest of pharmaceutical companies. OPCO (http://www.opko.com/, accessed on 17 April 2023), for example, is developing a CURNA platform that uses ASOs against NAT transcripts to inhibit the activity of pathogenic NATs [27].

Based on the cis regulation of as-lncRNAs, the functions of a large number of unidentified non-coding as-lncRNAs can be predicted according to the functions of the known adjacent protein-coding genes. Such functional predictions will help researchers better design experiments and study unknown as-lncRNAs, which is crucial to forming a comprehensive understanding of the function of noncoding genomes, gene expression regulation, and organism development. However, as-lncRNAs may demonstrate more than one mechanism of action to regulate transcriptional activation or inhibition of their target genes. For many as-lncRNAs, there may be new, undiscovered mechanisms. Therefore, future studies on as-lncRNAs are still necessary. Strengthening the research on as-lncRNAs in relation to cancer will help reveal the aetiology of tumour occurrence and development and help guide diagnosis and treatment in the future.

## Figures and Tables

**Figure 1 biomolecules-13-00684-f001:**
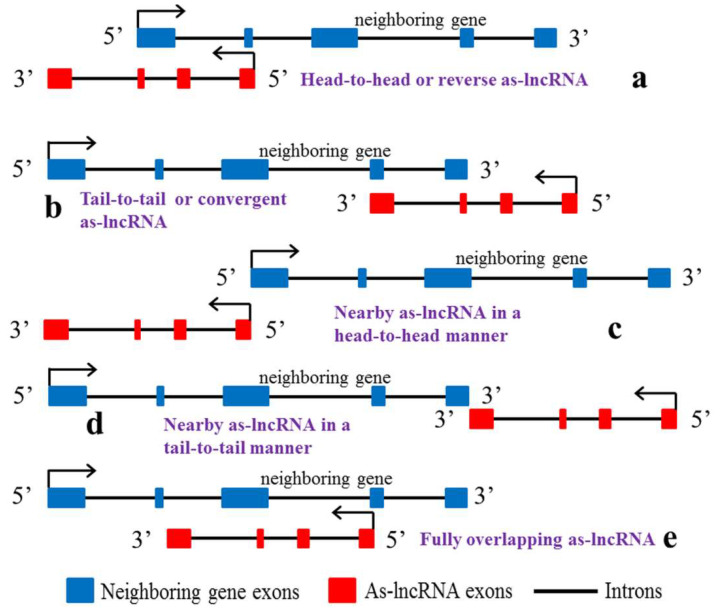
Classification of cis-acting as-lncRNAs. (**a**) Head-to-head as-lncRNAs, neighbouring genes and the 5′ ends of lncRNAs are aligned together. (**b**) Tail-to-tail as-lncRNAs, neighbouring genes are aligned with the 3′ end of lncRNAs. (**c**) Nearby as-lncRNAs in a head-to-head manner, the 5′ end of as-lncRNAs are near the 5′ end of the neighbouring genes. (**d**) Nearby as-lncRNAs in a tail-to-tail manner, the 3′ end of as-lncRNAs are near the 3′ end of the neighbouring genes. (**e**) Fully overlapping as-lncRNAs, as-lncRNAs completely overlap with neighbouring genes.

**Figure 2 biomolecules-13-00684-f002:**
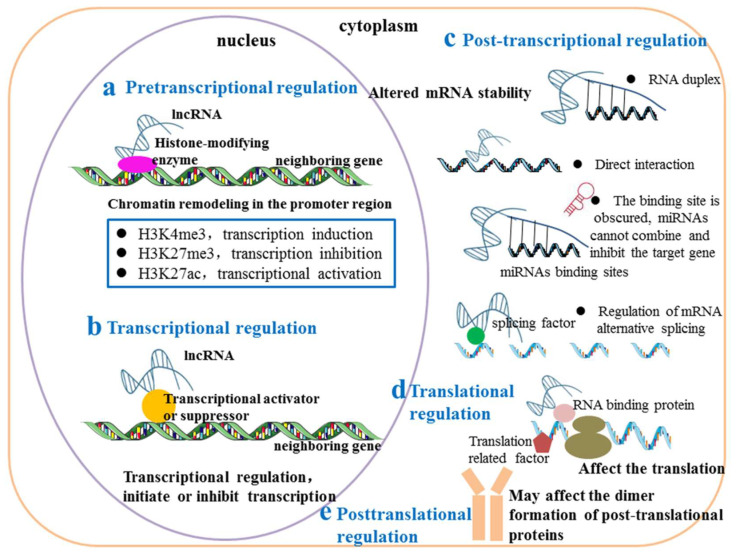
Regulation mechanisms of as-lncRNAs on neighbouring genes. (**a**) At the pretranscriptional level, as-lncRNAs reshape chromatin in the promoter region of neighbouring genes through histone modification enzymes, resulting in enrichment of H3K4me3/H3K27me3/H3K27ac in the promoter region, activating or inhibiting transcription of neighbouring genes. (**b**) At the transcriptional level, as-lncRNAs initiate or inhibit the transcription of neighbouring genes by interacting with transcription-related factors. (**c**) At the posttranscriptional level, as-lncRNAs regulate neighbouring gene expression by changing the stability of neighbouring gene mRNA or regulating the alternative splicing of neighbouring gene mRNA. as-lncRNAs can increase the stability of neighbouring gene mRNAs by forming RNA–RNA double strands with neighbouring gene mRNAs, change the mRNA stability of neighbouring genes by directly binding to the mRNA of neighbouring genes, or mask the binding site of miRNAs of neighbouring gene mRNAs to prevent the miRNAs from binding and inhibit the expression of neighbouring genes. (**d**) At the translation level, as-lncRNAs affect the translation of neighbouring genes by interacting with RNA binding proteins or translation-related factors. (**e**) At the posttranslational level, as-lncRNAs do not affect the expression of neighbouring genes but can affect the dimerization of neighbouring gene proteins, thereby affecting their activity.

**Table 1 biomolecules-13-00684-t001:** Regulation of trans-acting as-lncRNAs on target genes and their effects on tumor progression.

As-lncRNAs	Regulatory Mechanism	Influence on Tumour Progression
ZFPM2-AS1	Binds MIF and protects the stability of its protein	Promotes the progression of gastric cancer [16]
HOXD-AS1	Prevents the degradation of SOX4 mediated by miR-130a-3p	Facilitates liver cancer metastasis [17]
MNX1-AS1	Sponges miR-6785-5p and upregulates the expression of Bcl-2; inhibitesBTG2 expression through EZH2-induced H3K27me3 modification in the BTG2 promoter region	Contributes to gastric cancer progression [18]

**Table 2 biomolecules-13-00684-t002:** Effects of as-lncRNAs on the occurrence and development of different types of tumours by regulating neighbouring genes.

As-lncRNA	Regulatory Level	Regulatory Mechanism	Effects on Neighbouring Genes	Influence on Tumour Progression
ZNF667-AS1	Pretranscriptional level	Interacts with demethylase (TET1) and histone demethylase (UTX)	Transcription of ZNF667 is activated	Inhibits the progression of oesophageal squamous cell carcinoma [39]
IRF-AS	Transcriptional level	Forms a complex with RNA binding proteins ILF3 and DHX9	Transcription of IRF1 is activated	Inhibits the progression of oesophageal squamous cell carcinoma [41]
EZR-AS1	Pretranscriptional level	SMYD3 (lysine methyltransferase) is recruited and bound to the SBS-1 site in the EZR promoter region, causing H3K4 methylation in the EZR promoter region	Transcription of EZR is upregulated	Promotes the migration and invasiveness of oesophageal squamous cell carcinoma cells [33]
ZFPM2-AS1	post-transcriptional level	ZFPM2-AS1, ZFPM2 mRNA and UFP1 protein form a binding complex, which makes ZFPM2 mRNA unstable	Expression of ZFPM2 is downregulated	Promotes the proliferation, invasion, and EMT of lung adenocarcinoma cells [16]
TPT1-AS1	Transcriptional level	The transcriptional activity of TPT1 promoter is directly induced, but the degradation of TPT1 mRNA is not inhibited	Promotes the transcription of TPT1	Promotes proliferation, invasion, and metastasis of epithelial ovarian cancer [30]
SOCS2-AS1	Post-transcriptional level	Competitive a binding of miR-1264	Expression of SOCS2 is upregulated	Inhibits the proliferation and metastasis of colorectal cancer cells [59]
NR4A1AS	Post-transcriptional level	The direct binding of UPF1 protein to NR4A1 mRNA 3′UTR damages the binding of UPF1 protein to NR4A1 mRNA and prevents UPF1-mediated mRNA degradation	NR4A1 mRNA is more stable	Promotes the proliferation, migration, and invasion of colorectal cancer cells [44]
LDLRAD4-AS1	Post-transcriptional level	It directly interacts with LDLRAD4 mRNA and reduces the stability of LDLRAD4 mRNA mainly through its 1-1098 bp sequence region	Expression of LDLRAD4 is decreased	Promotes metastasis of colorectal cancer [48]
HOXD-AS1	Pretranscriptional level	Recruits PRC2 (histone methyltransferase) complex to bind to the promoter region of HOXD3 and induces the accumulation of inhibitory marker H3K27me3	Transcription of HOXD3 is inhibited	Inhibits the growth and metastasis of colorectal cancer [38]
PXN-AS1	Post-transcriptional level	The inclusion of exon 4 of PXN-AS1 is transcribed into a PXN-AS-L transcript containing exon 4 through splicing factor MBNL3	Promotes the expression of PXN mRNA and protein	Promotes the occurrence of liver cancer [58]
ID2-AS1	Pretranscriptional level	By blocking the binding of histone deacetylase 8 (HDAC8) to the ID2 enhancer, and increasing the accumulation of H3K27ac in the ID2 enhancer region	Promotes the transcription of ID2	Inhibits metastasis of liver cancer [40]
PTB-AS	Post-transcriptional level	By competitive combination the binding site of miR-9 in PTBP1-3 ‘UTR, miR-9 could not mediate negative regulation of PTBP1	The stability of PTBP1 mRNA is maintained	Promotes the occurrence of glioma [54]
PDCD4-AS1	Post-transcriptional level	Negatively regulates the binding of HuR to PDCD4 mRNA	Promotes the stability of PDCD4 mRNA	Inhibits the proliferation and migration of breast cancer cells [46]
MACC1-AS1	Post-transcriptional level	Promotes phosphorylation of AMPK, resulting in translocation of the RNA binding protein Lin28 from the nucleus to the cytoplasm	Enhances MACC1 mRNA stability	Promotes malignant phenotype of gastric cancer cells [49]
FGFR3-AS1	Post-transcriptional level	Forms RNA–RNA double strands with FGFR3 mRNA	Makes FGFR3 mRNA more stable	Promotes the progression of osteosarcoma [43]
IGFBP7-AS1	Post-transcriptional level	Forms RNA–RNA double strands with IGFBP7 mRNA	Makes IGFBP7 mRNA more stable	Inhibits the progression B-cell lymphoma [42]
ZEB1-AS1	Translation level	Recruits AUF1 and activates the translation of ZEB1 mRNA	Activates the translation of ZEB1 mRNA	Promotes bladder cancer cells migration and invasion [62]
IDH1-AS1	Posttranslational level	Alters IDH1enzyme activity	Enhances the enzyme activity of IDH1	Promotes prostate cancer progression [63]

## Data Availability

Not applicable.

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
