# Peer review of "The Roles of Antisense Long Noncoding RNAs in Tumorigenesis and Development through Cis-Regulation of Neighbouring Genes"

_biomolecules, 2023, doi:10.3390/biom13040684_

Round 1
Reviewer 1 Report
Binyuan Jiang and co-workers provide an update on the regulatory role of natural antisense transcripts (NATs) with particular focus on cancer development. They introduce the nomenclature of NATs, discuss cellular mechanisms triggered by the expression of NATs and follow up on examples of NATs with links to cancer. Updates in the field of NATs are welcome, though, there are considerable concerns with the submitted paper that need to be addressed.
- The term ‘as lncRNA’ is confusing because it can be read as ‘as lncRNA as well as ‘as’ ‘lncRNA’. Why not use as-lncRNA or stick with NAT?
- Traditionally, NATs have been categorized as head-to-head, tail-to-tail and fully overlapping, with additional specifications such as ‘intronic’ or ‘non-overlapping’. The term ‘nearby’ is confusing and vague and should be avoided.
- The introductory paragraph to section 3 needs major revisions. The statement ‘higher cell and tissue specificity’ may be correct for lncRNAs- though NATs are generally co-expressed with their sense transcripts (https://genome.cshlp.org/content/13/6b/1324.full.html and many other reports). The statement that there are only two alleles of a gene and hence only two molecules of RNA are needed to regulate them is too blunt. NATs are unlikely to base-pair directly with genomic DNA in a targeted, regulatory way, hence there will be higher levels of RNA required to make a regulatory impact. These may still be low, but probably much higher than just two molecules.
- The concept of ‘pretranscriptional’ level of regulation is problematic -and in a sense a ‘chicken end egg’ question. The placement of epigenetic marks generally requires transcriptional activity, even if the outcome is (re)silencing of a gene. The contribution of NATs to such epigenetic mechanism can therefore be considered as transcriptional and the latter term may be used rather than the rarely used ‘pre-transcriptional’.
- The selection of NATs involved in cancer appears random and is by no means comprehensive, for good reasons, there are simply too many to be all listed in such a review. Though, a useful resource and add-on to this review would be a searchable database detailing the features of human NATs as in table 1.
- The formation of double-stranded RNA between sense and dsRNA is mentioned. However, with dsRNA formation comes the risk of PKR/MDA5 activation and innate signalling. How is this mitigated, is it low levels of dsRNA or downregulated dsRNA sensors or are there other options?
- Generally, English grammar and spelling are fine, however, there are a few odd wordings that need attention:
line 12 ‘is a reverse transcribed lncRNA’ -> ‘is a lncRNA transcribed in reverse orientation’
line 82 ‘compared with encoding genes’ -> ‘compared with protein encoding genes’
line 87 ‘... its antisense splicing form’ ?
lines 115,171 ‘RNA double chain’ -> ‘RNA double strand’
line 185 ‘3’-UTR damages the binding…’ -> ‘3’-UTR inhibits/reduces the binding…’
line 215 from collecting polyribosomes -> from assembling polyribosomes
table 1: The description of the regulatory mechanism needs correcting: ZFPM-2-AS1 (binding); SOCS2-AS1 (adsorption -> binding); LDLRAD4-AS1 (destroys -> reduces, its 1-1098 bp sequence region ????); PTB-AS (rephrase the description)
line 284 aroused ->attracted
Reviewer 2 Report
Review on the manuscript titled “The roles of antisense long noncoding RNAs in tumorigenesis
and development through cis-regulation of neighboring genes” by Jiang et al., 2023
The authors present a review of as-lncRNA effects on the host genes featuring the range of elucidated cases compiled in Table 1. Notably, the authors primarily targeted tumor –associated lncRNA cases. As could be seen from Table1, the major instances listed maintain posttranscriptional level. (13 of 16 instances overall).
While the compilation presented should render an interest for the researchers in the field of cancer, there is a list of comments that authors should address.
1) AS-RNA cannot encompass 50-70% of total lncRNA pool as mentioned by authors: the sense lncRNAs quantity is at least twice as much, not to mention pseudogenes, intergenic lincRNA. I didn’t find in [3] the reference for this amount.
2) The authors didn’t mention that AS-RNA are highly enriched in CpG islands promoters genes as published before.
3) S161 “covering the binding sites of miRNAs,” – blocking?
4) The authors outline 5 stages of as-lncRNA interference in 3.1 section, but the table 1 comprises only 3 cases. I can’t believe there are no cancer-associated instances for the rest of two ones.
5) “Trans-acting as lncRNAs” category should be represented (maybe as an additional field or as an instance) in the table 1.
6) Figure 1 should be supplied with the as-lncRNA entity proportions (let it be eve a rough one) in each case in additional/inserted table.
7) Concerning the basic “Transcriptional collision” phenomenon: I’ve frequently observed the co-expression of both host and AS-lncRNA genes in expression data. It was more frequent than antagonistic expression. Maybe there’s an explanation that would be quite relevant in the review, please address it.
8) English should be check spelled.
Round 2
Reviewer 1 Report
Thank you for revising the manuscript and rebutting the issues raised. Most of the concerns have been delt with satisfactorily. Two open questions remain, concerning points 5 and 6 in the rebuttal letter.
5) The authors have misunderstood my point. All the databases that are listed and discussed in the revised manuscript are no longer active. To my knowledge, there are no reasonably updated sites specifically for antisense transcripts. Such a repository would need to be generated -or an old one revived. Of course, such a task exceeds the scope of a simple review article, though the authors may include promote the idea to generate one.
Please remove ‘5. Databases about Natural Antisense Transcripts’ and Table 3.
6) The authors have not answered the question. The cited paper describes a viral protein that mitigates dsRNA after viral infection, though as-lncRNAs are not related to a viral challenge. Do the authors suggest a cellular homologue of the viral C protein? Or have I missed something?
Author Response
Reviewer #1: Thank you for revising the manuscript and rebutting the issues raised. Most of the concerns have been delt with satisfactorily. Two open questions remain, concerning points 5 and 6 in the rebuttal letter.
5) The authors have misunderstood my point. All the databases that are listed and discussed in the revised manuscript are no longer active. To my knowledge, there are no reasonably updated sites specifically for antisense transcripts. Such a repository would need to be generated -or an old one revived. Of course, such a task exceeds the scope of a simple review article, though the authors may include promote the idea to generate one.
Please remove ‘5. Databases about Natural Antisense Transcripts’ and Table 3.
Responses:Many thanks for the reviewer's suggestion, we have removed ‘5. Databases about Natural Antisense Transcripts’ and Table 3. Please see the revised manuscript 1.
6) The authors have not answered the question. The cited paper describes a viral protein that mitigates dsRNA after viral infection, though as-lncRNAs are not related to a viral challenge. Do the authors suggest a cellular homologue of the viral C protein? Or have I missed something?
Responses:If the reviewer mean this article”IGFBP7-AS1 is a p53-responsive long noncoding RNA downregulated by Epstein-Barr virus that contributes to viral tumorigenesis”? EBV can reduce the formation of double-stranded mRNA of IGFBP7-AS1 and IGFBP7 through some regulatory mechanisms, thus reducing the expression of tumor suppressor protein IGFBP7, and thus promoting the occurrence and development of B-cell lymphoma. This paper reveals a new mechanism by which EBV promote tumor progression by regulating host LncRNA IGFBP7-AS1. AS to “Do the authors suggest a cellular homologue of the viral C protein”, they did not touch upon it. Viral product double stranded RNA is different from “host double-stranded RNA”. As far as we know, it is unlikely that the virus C protein has cellular homologues. We don't know enough about the virus field.